



# Using Open-Path Dual-Comb Spectroscopy to Monitor Methane Emissions from Simulated Grazing Cattle

Chinthaka Weerasekara,[1] Lindsay C. Morris,[2] Nathan A. Malarich[3], Fabrizio R. Giorgetta,[3,4] Daniel I. Herman, [3,4] Kevin C. Cossel,[3] Nathan R. Newbury,[3] Clenton E. Owensby,[1] Stephen M. Welch,[1] Cosmin Blaga,[2] Brett D. DePaola,[2] Ian Coddington,[3] Eduardo A. Santos,[1] and Brian R. Washburn[3]

[1]Kansas State University, Department of Agronomy, Manhattan, KS, 66506, United States
[2]Kansas State University, Department of Physics, Manhattan, KS, 66506, United States
[3]National Institute of Standards and Technology, Communications Technology Laboratory, Boulder, CO, 80305, United States
[4]University of Colorado, Boulder, Department of Physics, Boulder, CO, 80309, United States

*Correspondence to*: Brian R. Washburn (brian.washburn@nist.gov), Eduardo Santos (esantos@ksu.edu )

**Abstract.** Accurate whole-farm or herd-level measurements of livestock methane emissions are necessary for anthropogenic greenhouse gas inventories and to evaluate mitigation strategies. A controlled methane ($CH_4$) release experiment was performed to determine if dual comb spectroscopy (DCS) can detect $CH_4$ concentration enhancements produced by a typical herd of beef cattle in an extensive grazing system. Open-path DCS was used to measure downwind and upwind $CH_4$ concentrations from ten point-sources of methane simulating cattle emissions. The $CH_4$ mixing ratio along with wind velocity data were used to calculate $CH_4$ flux using an inverse dispersion model, and the simulated fluxes were then compared to the actual $CH_4$ release rate. For a source located 60 m from the downwind path, the DCS system detected 10 nmol mol$^{-1}$ $CH_4$ horizontal concentration gradient above the atmospheric background concentration with a precision of 6 nmol mol$^{-1}$ in 15-min interval. A $CH_4$ release of 3970 g day$^{-1}$ was performed resulting in an average concentration enhancement of 24 nmol mol$^{-1}$ of $CH_4$. The calculated $CH_4$ flux was (4002±1498) g day$^{-1}$ in agreement with the actual release rate. Periodically altering the downwind path, which may be needed to track moving cattle, did not adversely affect the ability to determine the $CH_4$ flux. The measurement was only limited by maintaining sufficient reflected power from the remote retroreflectors over the open path to achieve a sufficient signal-to-noise ratio. These results give us confidence that $CH_4$ flux can be determined by grazing cattle with low disturbance and direct field-scale measurements.

## Introduction and motivation

Methane ($CH_4$) emissions from enteric fermentation in domestic ruminants is the largest anthropogenic source of $CH_4$ in the United States, with the dairy and beef industries being responsible for most of these emissions (EPA, 2023). Previous life-cycle analyses indicate that 70 to 80% of the total greenhouse gas (GHG) emissions from the beef sector occur during the grazing phase (Rotz et al., 2015; Thompson and Rowntree, 2020; Alemu et al., 2017). However, direct herd-scale $CH_4$ emission data in grazing systems are scarce. The low animal density and high animal mobility commonly found in most grazing systems



makes herd-scale measurements quite challenging (Felber et al., 2015; Dengel et al., 2011; Flesch et al., 2018; Laubach et al., 2016; Stoy et al., 2021). Accurate whole-farm and herd-level measurements of livestock methane emissions are necessary to evaluate mitigation strategies to reduce GHG emissions, improve current GHG national inventories and to assist governments, industries, and other organizations to fulfil commitments to reduce anthropogenic GHG emissions.

Methane emissions from individual animals have been measured using face masks (Place et al., 2011), head-hood chambers (Hill et al., 2016), whole-animal respiration chambers (Pinares-Patiño et al., 2011), tunnels (Lockyer and Jarvis, 1995), automated spot head-box measurements (Hristov et al., 2015) and tracer methods (Grainger et al., 2007; Johnson et al., 1994). The respiration chamber is considered the standard technique for measuring livestock $CH_4$ emissions. Results from chamber studies have been used to develop predictive models and equations for national GHG inventories (Danielsson et al., 2017; Ramin and Huhtanen, 2013). However, chambers can create measurement artefacts by affecting animal behaviour and are not practical for measuring $CH_4$ emissions from many animals (Storm et al., 2012).

Micrometeorological techniques have been applied for measuring ammonia, carbon dioxide, nitrous oxide and $CH_4$ emissions from livestock systems (McGinn and Flesch, 2018b; Phillips et al., 2007; Sun et al., 2015; Prajapati and Santos, 2018b; Laubach et al., 2024), and have the advantages of being non-intrusive, can integrate fluxes from large areas or herds of cattle reducing measurement uncertainties due to animal-to-animal variability, and provide high temporal resolution (<1 h) flux measurements (McGinn, 2013). The widely used eddy covariance technique has been combined with flux footprint models to estimate methane emissions from ruminant herds (Coates et al., 2017; Prajapati and Santos, 2018a; Dengel et al., 2011; Stoy et al., 2021). However, this approach requires that the presence of animals in the flux tower footprint, which makes its implementation challenging in extensive grazing systems where cattle often do not remain for long periods in the area sampled by the flux tower.

Lagrangian stochastic models, which are the basis for several inverse dispersion models (IDM), have been used to infer emissions of gases such as ammonia and $CH_4$ from agricultural systems (McGinn and Flesch, 2018b; Flesch et al., 2005; Laubach and Kelliher, 2005). Unlikely traditional micrometeorological methods, such as the eddy covariance and flux gradient methods, they can handle source areas of different sizes and complex source geometries (Flesch et al., 2005). The IDM proposed by Flesch et al. (1995) has been used to quantify $CH_4$ emissions from ruminants (Laubach and Kelliher, 2005; Flesch et al., 2018; McGinn et al., 2011; Prajapati and Santos, 2018a). In typical IDM applications, open-path Fourier Transform Infrared (FTIR) sensors are setup upwind and downwind for the source of interest. The gas emission rates are then inferred based on the increase of gas concentration downwind from the source and turbulence statistics obtained from wind velocity measurements. McGinn et al. (2011) used IDM to estimate methane emissions from 18 animals grazing in a 1-ha paddock. They measured the area with five different paths ranging from 80 m to 128 m in length, so that at least one laser path was close enough to the cattle for their open-path FTIR system to be able to detect an enhancement in concentration. The limited path distance, bulky apparatus, multi-component retroreflectors (Bai et al., 2022) make employing open-path FTIR challenging in agricultural environments.



## Dual comb spectroscopy

Dual-comb spectroscopy (DCS) is a spectroscopic technique that uses two coherent frequency combs to get molecular concentrations through absorption (Coddington et al., 2016). A frequency comb is a laser spectrum composed of many ($10^6$) regularly spaced (MHz) spectral lines known as comb teeth with spectral coverage of multiple THz. Two frequency combs with slightly different repetition rates pass through a gas. Atmospheric molecular absorption lines, such as those due to $CH_4$, have GHz-wide absorption features and will absorb multiple comb teeth. After passing through the gas, the light from the combs is incident onto a square-law photodetector generating a radio frequency (RF) comb composed of heterodyne beats between pairs of optical comb teeth. From this an electrical interferogram (IGM) is generated, and its Fourier transform provides both the gas absorption and laser spectra. DCS is a sensing tool that combines and enhances the most desirable traits of FTIR and tunable diode laser absorption spectroscopy to measure entire absorption bands of multiple gas species at high speed with fine spectral resolution. In particular, DCS offers the unique ability to interrogate kilometer-scale paths and reliably measure very small changes in gas concentration making DCS ideal for quantifying fluxes of agriculturally significant gases in the field scale.

DCS is commonly used in an open-path differential measurement geometry to measure gas mole fraction on two beam paths to determine $CH_4$ flux from a source area. As seen in Fig. 1a, comb light generated from the DCS system in a trailer is split and sent on upwind and downwind paths. A sample IGM (Fig. 1b), from each path is recorded and its Fourier transform provides both the gas absorption and laser spectra (Fig. 1c). In order to obtain gas mole fraction, the spectral absorption is fit using a nonlinear curve fitting routine (Newville, Matthew; Stensitzki, Till; Allen, Daniel B.; Ingargiola, Antonino, n.d.) using molecular information from the HITRAN spectral database (Gordon et al., 2017; Rothman et al., 2009). The open-path DCS system used for this study has spectral coverage from 179.8 THz to 188.9 THz (6000 cm$^{-1}$ to 6300 cm$^{-1}$) and with a spectral resolution of 200.005 MHz (0.00667 cm$^{-1}$). The system is designed to target $CH_4$, $CO_2$, and water vapor with laboratory-level precision while operating in the field. It is based on all-polarization-maintaining, mode-locked erbium-doped fiber lasers with repetition frequencies 200,005,000 Hz and 200,005,000 + 208.88 Hz respectively (Sinclair et al., 2015). Mutual comb coherence is established by phase-locking each comb to the same free-running continuous wave laser at 192.175 THz and by phase-locking the carrier-envelope offset frequency of each comb using an in-line $f$-to-$2f$ interferometer (Truong et al., 2016). To tailor the comb spectrum to cover the $CH_4$ absorption band at 181.97 THz, light for each comb is amplified in an erbium-doped fiber amplifier and sent through a short piece of highly nonlinear fiber. For the DCS measurement, the filtered outputs are combined using a fiber combiner generating two outputs that are directed over two open-air paths.

Each IGM is digitally sampled with 14 bits and contains 957500 points. The IGMs are generated at a rate of 208.88 Hz, so streaming and storing these data to a computer would require terabits of storage. To reduce data storage requirements, during the course of the measurement, 28 IGM's are co-added by a field programmable gate array (FPGA) to produce a hardware-averaged IGM. These IGMs are streamed to a computer, which performs phase-correction and additional

averaging using methodology similar to techniques used in FTIR  (Griffiths and de Haseth, 2006).  The computer calculates a

phase-corrected IGM every 5 min and stores it in the hard drive. For the best case, 2238 hardware-averaged IGMs are used to

generate a phase-corrected IGM every 5 min.    Hardware-averaged IGMs with poor SNR, mostly due to poor alignment

between transceiver and the retroreflector, are rejected and not used in the phase-correction. Under moderate windy conditions

IGM rejection is less than 10%.

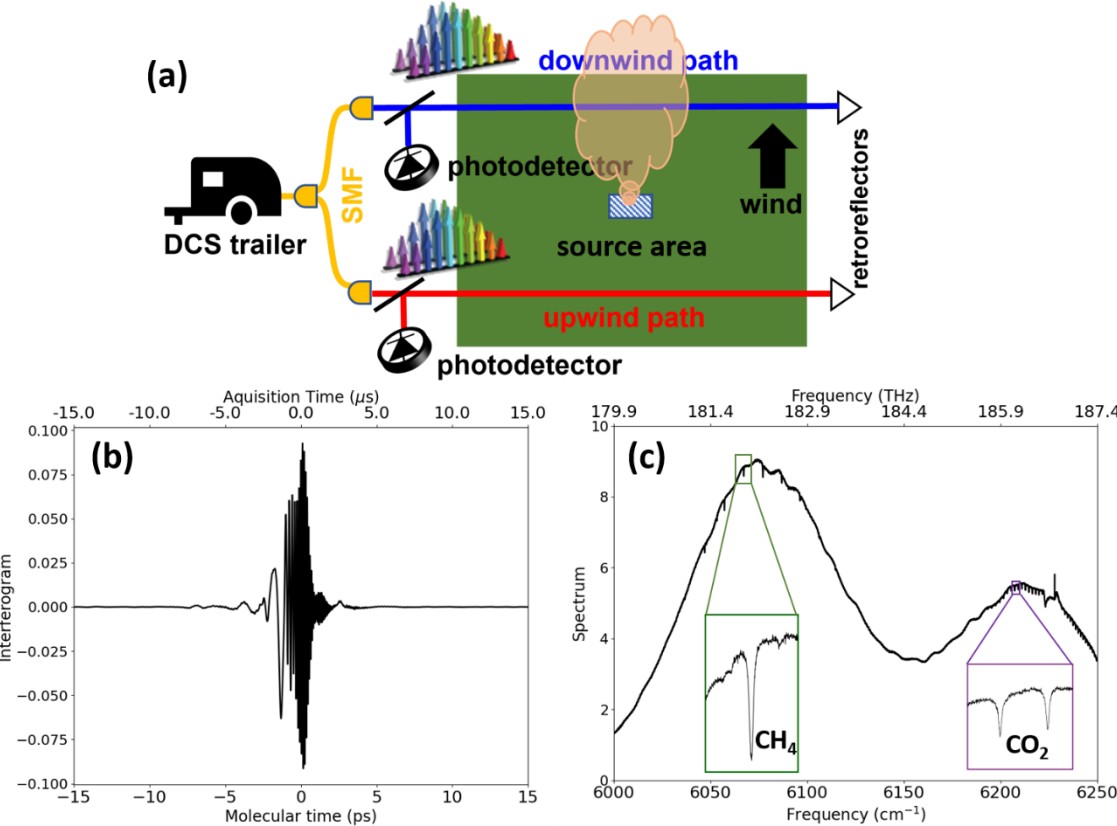

**Figure 1: (a) Schematic of the dual-comb spectrometer gas concentration measurements on two paths from a CH₄ source area.**
**Yellow lines indicate single-mode fiber (SMF) transmitting dual-comb light to an upwind (red) and downwind (blue) open-air paths.**
**CH₄ is emitted from an area between the two paths under proper wind directions. RF signals from two photodetectors are sent back**
**to the trailer and two interferograms (IGM) containing gas concentration information for each path are digitized. (b) A dual-comb**
**spectroscopy phase-corrected IGM after 5 min acquisition time on the upwind path.  'Acquisition time' is the microsecond timescale**
**of the measured RF voltage that directly corresponds to picosecond molecular motion of 'Molecular time' (c) The Fourier transform**
**of the IGM with insets showing CH₄ and CO₂ absorption lines and the laser baseline.**

**Sensitivity and precision required for grazing measurements**

A forward Lagrangian stochastic model (Windtrax, Thunderbeach Sci.) was used to simulate the concentration field downwind

from a hypothetical herd of 20 head of beef cattle grazing in an area of 25 ha, which is a typical stocking density (animal/area)

in the Flint Hills region, Kansas (Fig. 2). Turbulence data for the simulations were measured using a sonic anemometer at a



grazing unit adjacent to the measurement site. The wind dataset selected for these simulations consisted of about 30 days in June, 2021 during the grazing season. To investigate if the DCS system can resolve the typical increase in mixing ratio above the typical $CH_4$ atmospheric background level (2000 nmol mol$^{-1}$), two expected $CH_4$ emission scenarios were valuated: 100 g head$^{-1}$ day$^{-1}$ and 300 g head$^{-1}$ day$^{-1}$. These values selected based on the reported IPCC Tier 1 emission values for grazing cattle in North America of 208 g head$^{-1}$ day$^{-1}$ (Eggleston et al., 2006). which is confirmed by a study of grazing 50 cow-calf pairs

(Todd, Richard W. et al., 2016).

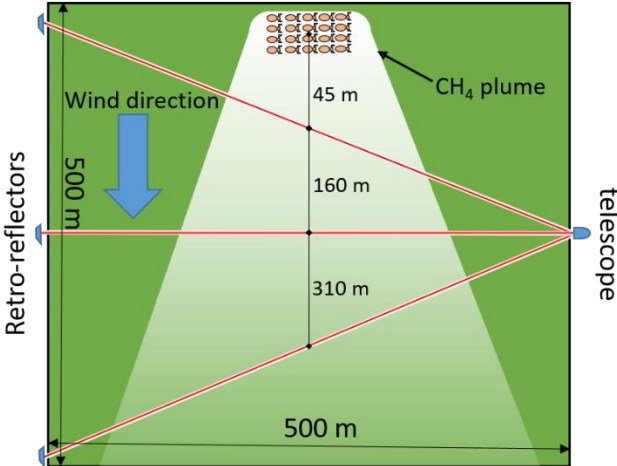

**Figure 2: Schematic diagram showing the location of the hypothetical herd of cattle, transceiver, retroreflectors and three possible downwind paths used for the forward WindTrax simulations. A constant background of 2000 nmol/mol was assumed so no upwind path was used in the simulation and not shown in the figure.**

The forward model predicted that a herd of 20 cattle grazing in an area of 25 ha would produce a $CH_4$ enhancement of 16 nmol mol$^{-1}$ above a 2000 nmol mol$^{-1}$ background for a beamline 45 m away from the herd of cattle assuming an emission rate of 300 g head$^{-1}$ day$^{-1}$ of $CH_4$. The enhancement drops to 2 nmol mol$^{-1}$ for a beam path 310 m away assuming the same emission rate. For a low emission scenario (100 g head$^{-1}$ day$^{-1}$), the $CH_4$ enhancements ranged from 5 to 1 nmol mol$^{-1}$ for a beam line located at 45 m and 310 m away from the center of the herd.


**Table 1 - Grazing system methane emission WindTrax simulation results showing the expected average $CH_4$ concentration measured by line sensors positioned downwind from a cattle herd with two $CH_4$ emission rates. The $CH_4$ background level was assumed to be constant at 2000 nmol mol$^{-1}$.**

| | Cattle $CH_4$ emission rate (g head$^{-1}$ day$^{-1}$) | | | | | |
|---|---|---|---|---|---|---|
| | 100 | | | 300 | | |
| **Distance (m)** | 45 | 160 | 310 | 45 | 160 | 310 |
| **[$CH_4$] (nmol mol$^{-1}$)** | 2005 | 2002 | 2001 | 2016 | 2006 | 2002 |



The small $CH_4$ enhancements shown in Table 1 will need to be measured by the DCS system in order to monitor emissions from a herd of cattle. Thus, it is important to consider the measurement's sensitivity and precision to $CH_4$ concentrations. The sensitivity of DCS to measure gas concentrations is proportional to the signal-to-noise ratio (SNR) and the number of absorption lines measured (Newbury et al., 2010). Experimentally the SNR is determined from the spectrum by dividing the peak signal intensity near 6074 $cm^{-1}$ by the average noise intensity in a 0.033 $cm^{-1}$ band centered at 5545 $cm^{-1}$.

Typical SNR values vary from 1000 to 3000 which is comparable to previous outdoor DCS measurements (Herman et al., 2021). The five strongest $CH_4$ absorptions lines (Fig. 3a) have a normalized absorption of roughly $-\exp(-aL) \approx aL = 0.03$, where $\alpha$ is absorption coefficient and $L$ is the path length. With a SNR of 3000 the measurement should be able to detect 5 nmol/mol enhancement above a 2000 nmol/mol background, specifically 2000 nmol $mol^{-1}*\frac{1}{3000}\frac{1}{5*0.03} \approx 5$ nmol $mol^{-1}$. The DCS concentration measurement precision under field conditions was determined using Allen-Werle analysis (Werle, 2011)

which includes effects of field-condition induced misalignment to the retroreflectors which causes fluctuations in the SNR. The result of an Allen-Werle analysis on a dataset taken for 24 hours on 12/18/2022 is shown in (Fig. 3b), showing a precision of 6 nmol $mol^{-1}$ $CH_4$ in 900 s (15 min) for 200-m paths. This result is consistent with results of (Herman et al., 2021) where data were taken with a SNR of 1000 and a precision of 25 nmol $mol^{-1}$ in 5 min.

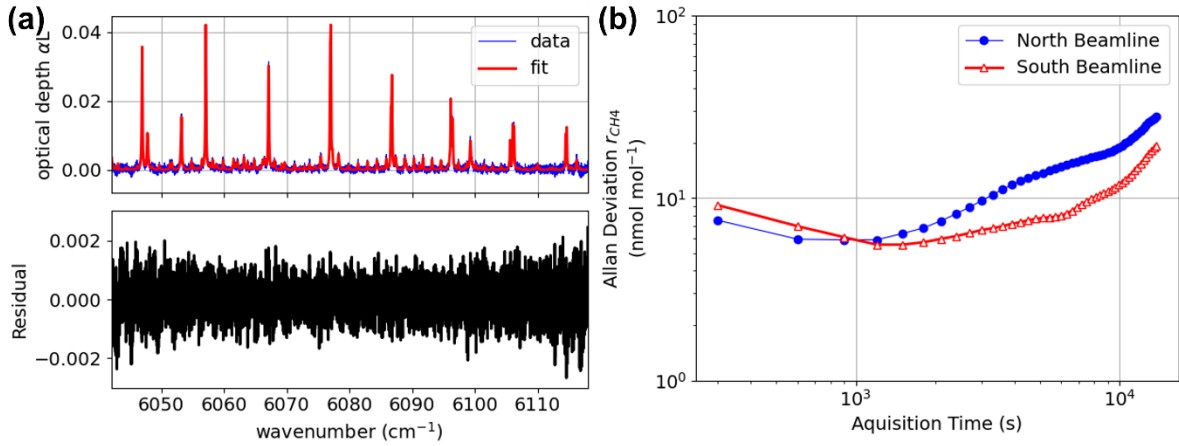

**Figure 3: (a) Result of cepstral-domain fitting of $H_2O$, $CH_4$, and $CO_2$ for 300 s averaged data, showing the resulting optical depth data, fit, and fit residual. (b) Allen-Werle deviation of the $CH_4$ mixing ratio ($r_{CH4}$) showing a 6 nmol $mol^{-1}$ $CH_4$ precision in 900 s.**

**Controlled $CH_4$ release experiment**

Controlled $CH_4$ release field experiments were conducted on the Rannells' Flint Hills Prairie Preserve (hereafter Rannells'

ranch) near Manhattan, Kansas USA ($39^o$ 08' 28''N, $96^o$ 31' 31''W, 324 m asl). The dominant steer grazing system in the Kansas Flint Hills is Intensive Early Stocking (IES) (Smith and Owensby, 1978). IES is a grazing system that takes advantage



of the early summer high quality forage by stocking at twice the normal season-long stocking rate (1.25 steers/ha) for the first
half of the growing season (~May 1 to ~July 15) with no grazing during the last half. The grazing unit used in the study has
31 hectares and has been annually burned in late April each year. This grazing unit was selected because its topography is
suitable for micrometeorological measurements and because it allows unobstructed paths for the DCS system for most of its
extension.

Previous work (Coburn et al., 2018; Alden et al., 2019) used DCS to measure simulated $CH_4$ leaks from oil and gas
production at the level of 1400 g day$^{-1}$ from a distance of 1 km. Here we seek to provide a similar verification of the technology
but with two important changes in the measurement configuration appropriate for livestock-based methane sources. First, the
sources will be distributed rather than concentrated to single point source, Secondly the sources are further from the
measurement paths. This larger separation will be necessary to accommodate the fact the herd will wander over time. The
measurement paths might be adjusted to accommodate the cattle movement but there will be a limit to how close the
measurement paths can be kept from the source. The main goal of this study is to determine if the DCS can detect small $CH_4$
concentration enhancements downwind from the area of interest, equivalent to those caused by a typical herd of cattle grazing
on an extensive pasture.

The DCS system was housed in a temperature-controlled trailer at the Rannells' Ranch as seen in Fig. 4. Single-mode
fibers (SMF-28) of lengths 10 and 40 m carried the dual-comb laser output light to two telescope transceivers (Fig. 4b) that
was used to send comb light across the North (blue) and South (red) beamlines. The transceiver consisted of an FC/APC fiber
termination followed by a collimating 179 mm focal length, 102 mm diameter, 45° off-axis parabolic mirror resulting in a
collimated beam of ~35 mm diameter. Eye-safe (<10 mW) collimated dual-comb light was directed with a 127 mm clear
aperture, 5 arcsec gold retroreflector (Edmund Optics[†]) positioned 200 m away (Fig. 4a) and the reflected signal was focused
onto a 150-MHz bandwidth photodetector (PDA10CF, Thorlabs) in the transceivers. RF signals from photodetectors were
transmitted to the trailer through RF cables (RG58, Pasternack) and digitized using a 14-bit digitizer (FMC104, Abaco).  To
remove any concentration bias due to digitizer nonlinearities we added a dither signal to the received DCS interferogram
(Malarich et al., 2023). The dither improved the individual channel precision by 5% and reduced the differences between
channels to below 3 nmol mol$^{-1}$.

Both transceivers were mounted on motorized tip/tilt gimbals (PT100, FLIR) that were automatically aligned using a
datalogger (CR1000x, Campbell Sci.) or personal computer algorithms to the retroreflectors based on the return DC signal
from the photodetector. The transceiver also housed a visible camera (BFLY-PGE-50A2M-CS, FLIR) to aid with alignment
and a consumer 5W 850-nm LED flashlight to allow the user to see the retroreflectors with the visible camera during nighttime.
The datalogger-controlled alignment system was able to maintain sufficient power back from the retroreflector to the

---

[†] Certain equipment or instruments are identified in this paper in order to specify the experimental procedure adequately. Such identification
is not intended to imply recommendation or endorsement of any product by NIST, nor is it intended to imply that the equipment identified
are necessarily the best available for the purpose.





transceiver in moderate wind conditions for over 24 hours. The positions of retroreflectors, manifold, sonic anemometer and transceiver were measured using a multi-band real-time kinematic positioning (RTK) receiver (Reach RS2+, Emlid) with 7 mm and 14 mm horizontal and vertical accuracies, respectively. The horizontal and vertical coordinates obtained for the

transceivers and retroreflectors were then used to determine the path lengths shown in Fig. 4.

A custom-built gas manifold (Fig. 4c) was used to control the release of $CH_4$ through 10 point sources located within the two DCS beam lines. Methane gas from a compressed tank (99.97% purity) was delivered to a proportional solenoid valve (PVQ13, SMC, Noblesville, IN) using a two-stage pressure regulator and high-density polyethylene tubing (I.D. 5.3 mm). The proportional valve was then connected to a multi-port aluminium manifold using high density polyethylene tubing. The

pressure inside the manifold was monitored using a pressure transducer (PX119-030GI, Omega, Norwaok, CT). The $CH_4$ from the manifold flowed through ten 0.254-mm precision orifice assemblies (K2-10-SS, O'Keefe Controls Co., Monroe, CT). The precision orifice assemblies were then connected to 8-m high density polyethylene tubing lengths. The other extremity of these plastic tubes was then attached to metal rods at a height of 0.7 m above the ground. During $CH_4$ controlled-release campaigns, the pressure inside the manifold was adjusted to provide the desired flow rate by controlling the voltage applied to the

proportional valve using a datalogger (CR1000, Campbell Sci.). A feedback loop between proportional valve and pressure transducer ensured a constant pressure inside the manifold during the control release campaigns. The $CH_4$ tank was weighted in the beginning and end of the gas release campaigns and the mass of gas released was determined gravimetrically using a scale (D125WQL, Ohaus, Parsippany, NJ).

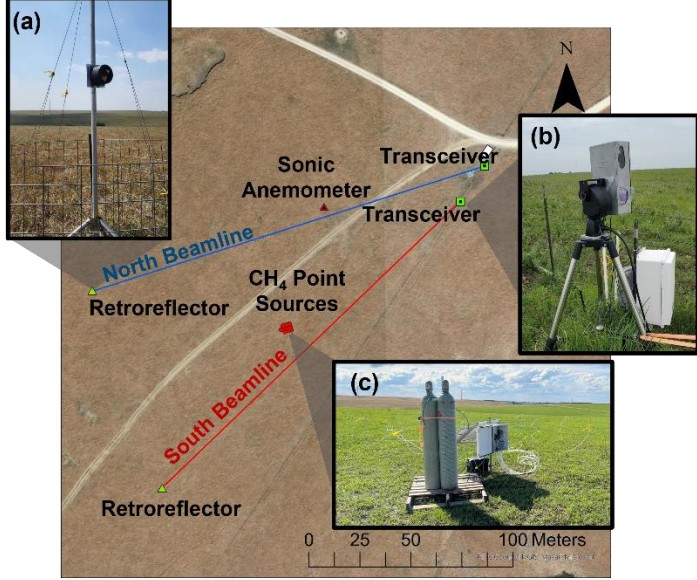

**Figure 4: Layout of the experimental site the Rannells' Flint Hills Prairie Preserve. Insets: (a) hollow gold retroreflectors, (b) optical transceiver on a tip/tilt gimbal, and (c) gas manifold and point sources used to release $CH_4$ at a known rate.**



### Obtaining CH₄ mixing ratio using spectral line fitting

Mole fractions ($\chi$) of $CO_2$, $H_2O$, and $CH_4$ were obtained from the measured interferogram using a fit model derived from a combination of the HITRAN databases (Rothman et al., 2013) and the cepstral-domain technique (Cole et al., 2019).

Temperature and pressure data used as an initial guess for the fit were provided by the sonic anemometer (CSAT3) and a pressure transducer (CS100, Campbell Sci.), respectively, which were both located on the same tower during the measurement campaign. The spectral band used in the cepstral-domain fitting was from 6000 cm$^{-1}$ to 6300 cm$^{-1}$ and contains $CH_4$, $H_2O$ as well as weak $CO_2$ absorption lines. The $CH_4$ spectral model (Rothman et al., 2009) contains multiple features with line-strengths $S$ greater than $10^{-22}$ cm$^{-1}$ molecule$^{-1}$ cm$^2$). A cepstral-domain filter operates in the time domain and removes broad

comb baseline structure in the IGM at times shorter than 15 ps and removes an etalon feature from 30 to 40 ps. The conversion from $CH_4$ mole fraction ($\chi_{CH4}$) to mixing ratio ($r_{CH4}$) was calculated using the fit $H_2O$ mole fraction ($\chi_{H2O}$) and

$$r_{CH4} = \frac{\chi_{CH4}}{1-\chi_{H2O}} \tag{1}$$

### Computing CH₄ flux using an inverse dispersion model

A freely available IDM software, WindTrax (Crenna, 2006), was used for computing $CH_4$ fluxes. The WindTrax input data consisted of measured upwind and downwind $CH_4$ mixing ratios, and appropriate wind statistics. As WindTrax flux estimates are more precise for 15 minute or longer timescales (Flesch et al., 2004), we averaged the 5-min DCS mole fraction data to 15-min. WindTrax requires appropriate weather conditions to provide accurate estimate of fluxes, so the data were screened based on the following acceptance criteria: wind friction velocity ($u^*$) > 0.1 m s$^{-1}$ and absolute Monin-Obukhov Length values

$|L_{MO}|$ > 10 m (Todd et al., 2014; Flesch et al., 2005). The source area (Fig. 4) used by WindTrax to infer fluxes was set to match the 12.5 m$^2$ area of the $CH_4$ point sources. In WindTrax all DCS measurement paths were modelled as line concentration sensors consisting of 60 particle "release" points along the path, starting at the transceiver and ending at the retroreflector.

One of the principal sources of uncertainty in the IDM estimates arises from the errors in the gas concentration

measurements themselves. The flux is dependent on the difference between downwind ($r_d$) and upwind ($r_u$) mixing ratios, measured the north and south beamlines (Fig. 4). The fractional uncertainty in the flux is given by:

$$\frac{\sigma_F}{F} = \frac{\sqrt{\sigma_{r_d}^2 + \sigma_{r_u}^2 - 2cov(r_d,r_u)}}{r_d - r_u} \tag{2}$$

where $F$ is the flux, $\sigma_F$ is flux error, $\sigma_{r_d}^2$ is downwind (background) mixing ratio error, $\sigma_{r_u}^2$ is upwind mixing ratio error, $cov(d,u)$ is the covariance of the downwind and upwind errors. A covariance term was added to the quadrature error following





previous studies (Herman et al., 2021; Bai et al., 2022) to account for small correlations in the different path errors. The mixing

ratio errors and the covariance were determined from the recorded 5-min measured SNR assuming that the mixing ratio error

is inversely proportional to the SNR. The fractional uncertainty ignores errors due to measurement deadtime, wind field

measurements and IDM inherent uncertainties (Flesch et al., 2004).  Typical values fractional uncertainty in the flux vary from

20 to 30%.

240                                  **Results from controlled CH$_4$ release measurements**

A CH$_4$ release at a rate of 3078 g day$^{-1}$ is shown in Fig. 5. CH$_4$ mixing ratio was measured at 5-min intervals for the North and

South laser beamlines as seen in Fig. 4. The CH$_4$ gas release started at 10:30 and ended at 18:00 on 24 Feb 2023. The mixing

ratio enhancement is given by $r_d - r_u$ (Fig. 5c). The small 10 nmol mol$^{-1}$ average enhancement can be seen fluctuating around

a 2026 nmol mol$^{-1}$ average background concentration. This measurement demonstrates that the DCS system can detect small

CH$_4$ enhancements equivalent to the ones caused by a small herd of cattle located at approximately 50 m from the downwind

laser beamline. The two-DCS measurement path geometry is also capable of capturing and rejecting the temporal dynamics of

the CH$_4$ background driven by changes in atmospheric boundary layer conditions.

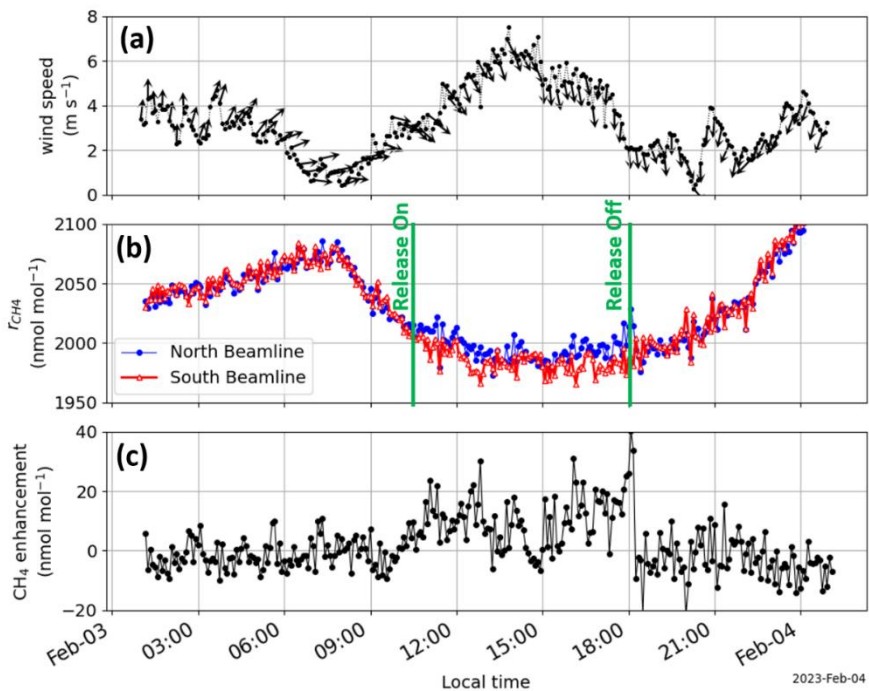

**Figure 5: 5-min values of (a) wind speed and direction, (b) CH$_4$ mixing ratio ($r_{CH4}$), and (c) enhancement during a controlled CH$_4$**
**release of 3078 g d$^{-1}$ equivalent to 15 head of cattle assuming a CH$_4$ rate of 208 g head$^{-1}$ day$^{-1}$. Wind arrows point in the direction**
**from which the wind is blowing. During the release, wind was mostly from the south causing an enhancement on the North Beamline.**

To determine if any mixing ratio biases exist between the North and South beamlines that may lead to incorrect flux values, north and south measurements were taken over 6.25 hours with no gas released with the wind from the west (Fig. 6). $CH_4$ mixing ratio and wind data were used to compute an average $CH_4$ flux of $(1\pm217)$ g day$^{-1}$ using the IDM. The average

$CH_4$ flux computed using WindTrax was $\pm974$ g day$^{-1}$ equivalent to approximately 5 head of cattle, assuming an emission rate of 200 g head$^{-1}$ day$^{-1}$.

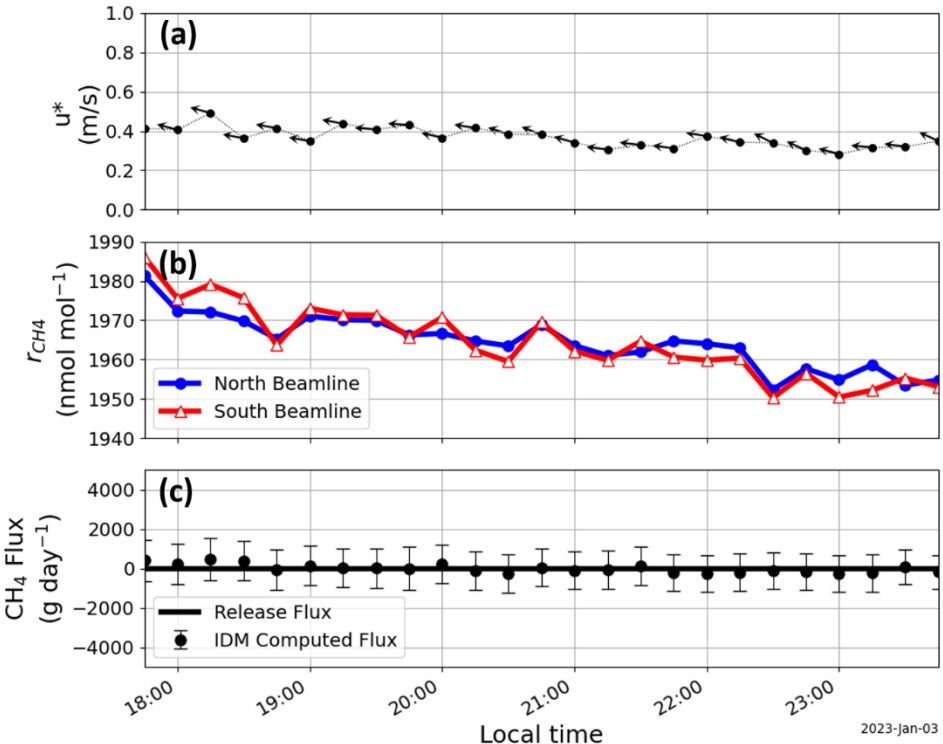

**Figure 6: Timeseries of (a) wind friction velocity and direction, (b) CH$_4$ mixing ratio ($r_{CH4}$), and (c) release and IDM computed CH$_4$ flux a case of no released gas.   The average CH$_4$ flux error is $\pm974$ g day$^{-1}$ or equivalent to $\pm5$ head of cattle. Wind arrows point in**
**the direction from which the wind is blowing.**

To test if the DCS measurement can be used correctly reproduce the release flux rate, a controlled $CH_4$ release corresponding to 3970 g day$^{-1}$, which simulates a 19-head cattle herd with emission rate of 200 g head$^{-1}$ day$^{-1}$, was performed where DCS measured concentrations and 3D wind statistics was measured for six hours (Fig. 7). DCS Mixing ratio and wind

data for 15-min intervals were then used to estimate $CH_4$ fluxes using WindTrax. The DCS system was able to detect the small 24 nmol mol$^{-1}$ average enhancement above the 2041 nmol mol$^{-1}$ average background concentration. WindTrax computed average $CH_4$ flux was $(4002\pm1498)$ g day$^{-1}$, showing a good agreement to the actual release $CH_4$ flux of 3970 g day$^{-1}$. The flux error bars were calculated using Equation (2).






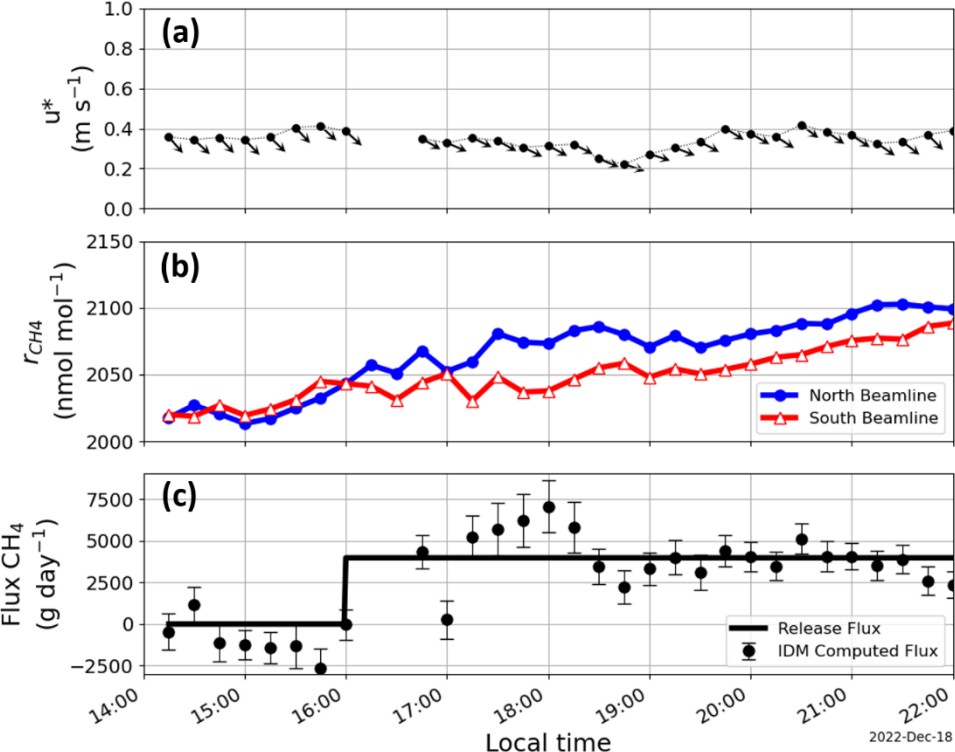

**Figure 7: Timeseries of (a) wind friction velocity and direction, (b) CH₄ mixing ratio ($r_{CH4}$), and (c) release and IDM computed CH₄ flux for a CH₄ release of 3970 g day⁻¹ equivalent to 19 head of cattle. Wind arrows point in the direction from which the wind is blowing.**

Monitoring grazing cattle emissions in the field will require changing between laser paths to capture emissions from a moving

herd. To investigate the effect of the distance between the herd and the beam paths, we alternated between two downwind

south paths (Fig. 8). Here a release simulating 40 head• was performed where the downwind south path was changed at hour

intervals during the release. Figure 8 shows the measured (a) wind conditions, (b) CH₄ mixing ratios, (c) release and IDM

computed CH₄ flux. During the measurement, the downwind path altered between south beamline 1 and 2 as a function of

time. The good comparison between the measured and calculated flux using both south beamlines demonstrated that altering

the beam paths did not adversely impact our ability to determine a flux from the source area.



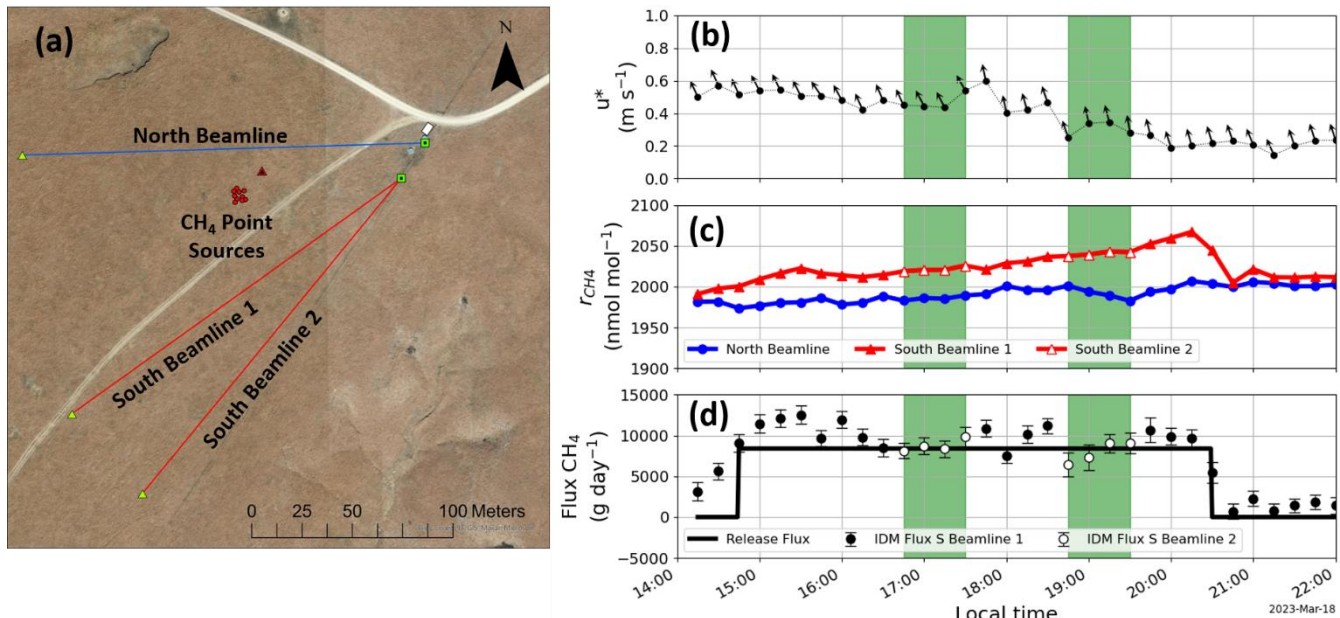

**Figure 8: (a) Layout of the experimental site the Rannells' Flint Hills Prairie Preserve used to alternate between two downwind south paths. Timeseries of (b) wind friction velocity and direction, (c) CH$_4$ mixing ratio, interferogram signal-to-noise, (d) release and IDM computed CH$_4$ flux for a release of 8396 g day$^{-1}$. The red triangle indicated the position of the 3D sonic anemometer. Wind arrows point in the direction from which the wind is blowing. The downwind South Beamline was changed during the release, focused on Retroreflector 2 at times 16:45 to 17:30, 18:45 to 19:30 (indicated by green shaded regions), and at Retroreflector 1 at all other times. Moving between the two downwind paths did not distort the concentration measurement or the computed IDM flux compared to the release rate.**

### Future work and conclusions

The agreement between the computed and actual CH$_4$ fluxes in this study shows that DCS can precisely measure the small CH$_4$ concentration enhancements due to a herd of beef cattle in the field at distances up to 100 m from the source area. Our results show that the DCS precision is mostly limited to the ability to maintain sufficient laser alignment between transceiver and retroreflector. Return power from the retroreflectors resulting in a SNR greater than 1000 was necessary to properly measure the small enhancements at the 6 nmol mol$^{-1}$ precision. A robust transceiver design and housing, along with a fast response datalogger-controlled gimbal alignment is critical to make continuous measurements under turbulent and varying environmental conditions.

In addition to the good precision, other important characteristics of the DCS measurement were highlighted in this study: 1) the use of inexpensive (US\$1.3 per meter) and robust telecommunication-grade fibre optics to transport the light from the DCS to outdoor transceivers over long distances (tens to hundreds of meters) with very low power losses (4.5% loss per km) and 2) its ability to measure multiple open atmospheric laser beamlines simultaneously with a single instrument. From a pure measurement standpoint, using a single instrument to measure gradients of concentration is desirable to eliminate measurement



biases. For example, prior calibrations are often necessary when using multiple FTIR systems to perform multi-path gas concentration measurements. The minimization of instrument biases is crucial when combining the DCS with existing micrometeorological techniques that utilize of vertical or horizontal gradients of concentration to infer fluxes (McGinn and Flesch, 2018a; Flesch et al., 1995). Expected $CH_4$ horizontal gradients in grazing systems are often small, as demonstrated in this study, so small instrument biases can lead to large errors when inferring fluxes. Furthermore, the use of a single instrument to measure multiple source areas will also lead to a reduction of the cost necessary to evaluate multiple treatments. This is particularly important when assessing GHG mitigation strategies, which often require evaluation of multiple treatments and management practices simultaneously.

The driving rationale of this work is to quantify the net $CH_4$ fluxes produced by cattle grazing system, which will require measuring 3D wind statistics and $CH_4$ concentration enhancements upwind and downwind of the animals over long times. Although soil $CH_4$ fluxes are expected to be smaller than animal emissions, they could be important for estimating whole-system $CH_4$ budgets. Separating animal and soil contributions to the net $CH_4$ fluxes will require a combination of measuring approaches, such as chamber and micrometeorological measurements (*e.g.* eddy covariance measurements). High animal mobility in extensive grazing systems will also pose additional challenges for the quantification of cattle emissions. The ability to track cattle for laser-based greenhouse gas detection is an open and significant problem. Animal tracking using GPS collars (Felber et al., 2015) or digital photographs (Stoy et al., 2021) have been used to track ruminants in grazing systems. Both approaches have their own challenges, GPS collars need to provide high accuracy and temporal resolution spatial data while consuming low power to allow animals to be monitored during an entire grazing season. Wide-angle camera images were used to determine the position of the cattle herd during Summer 2023 with limited success (data not shown), since it was difficult to properly discern animal positions with enough spatial resolution needed for the IDM. Ideally, real-time animal tracking using GPS collars, digital images or a combination of both could be used to improve flux estimate accuracies. This could be done by subdividing the grazing system into small monitoring areas. The area monitored by the DCS system could be selected by aiming the laser beam at different retroreflectors installed at different points of the pasture. By monitoring these small areas, it would be possible to keep the downwind laser beam close to the animals, thus, measuring a larger $CH_4$ concentration enhancement and reducing the uncertainties in concentration measurements. The ability of DCS to measure gases from a large area continuously will permit monitoring of $CH_4$ emissions from a slow-moving herd of cattle, providing precise $CH_4$ flux values to improve agricultural GHG inventories and management practices.

**Author contribution**

I.C., B.D.D, K.C.C, S.M.W., C.B, C.E.O, N.R.N., E.A.S., and B.R.W. conceived of and designed the experiments. E.A.S., D.I.H., N.A.M and B.R.W. built the DCS system. F.R.G and B.R.W wrote DCS data acquisition code. N.A.M and B.R.W. wrote the spectral line fitting code. C.W., L.C.M., E.A.S., and B.R.W. performed the experiments and analysed the data. C.E.O



is the manager of the Rannells' Ranch. L.C.M., C. W., K.C.C., E.A.S., and B.R.W. performed the experiments and analyzed
the DCS data. C.W. and E.A.S. implemented the inverse dispersion model. B.R.W. and E.A.S. supervised the project.

## Competing interests

The authors declare that they have no competing interests.

## Data and materials availability

Data associated with this paper is publicly available at https://doi.org/10.18434/mds2-3139 or can be obtained from the authors
upon a reasonable request.

## Acknowledgements

This work was partially funded by the NSF Division of Biological Infrastructure Award #1726304, the William and Joan
Porter Endowment, the Habiger Heritage Fund, and NIST.

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
