# Peer review of "Using Open-Path Dual-Comb Spectroscopy to Monitor Methane Emissions from Simulated Grazing Cattle"

_EGUsphere, 2024_

## Author Comment (AC1)

**Response to RC1's comments**

We would like to thank Reviewer 1 for the careful reading of our manuscript and the clear and thoughtful comments. Below we address each comment specifically. The reviewer comments are written out in **black** text. Our responses are indicated in **blue** font, with text added to the manuscript indicated in **red** font.

Review of *"Using Open-Path Dual-Comb Spectroscopy to Monitor Methane Emissions from Simulated Grazing Cattle"* **by Weerasekara et al., submitted to Atmospheric Measurement Techniques**

**General Comments**

This manuscript examines the potential of a line-averaging gas sensor (based on dual comb spectroscopy, DCS) for use in calculating methane ($CH_4$) emissions from grazing cattle. This is an important topic as grazing cattle are significant $CH_4$ emitters, and a sensitive and robust $CH_4$ sensor is needed to measure emissions in that environment. The subject is appropriate for Atmospheric Measurement Techniques.

The paper is interesting but there is a lot going on in the short manuscript. What are the objectives of the paper: a modelling study to estimate $CH_4$ enhancement levels downwind of cattle; a gas release trial to determine the enhancement; an examination of DCS sensitivity relative to the cattle signal; an examination of the accuracy of the IDM technique for calculating emissions? The manuscript touches on all of these, but generally without enough description and discussion to address each adequately. The authors likely intend this work as giving a short-overview of the potential of DCS, but it becomes an overly simplified paper that tries to do too much. My main recommendation to the authors is to revise the manuscript to follow a simpler objective(s), so that an expanded explanation and discussion can be added (for the more focused objective). Along these lines:

1.    There should be a clear objectives statement in the introduction.   There is such a statement deep in the manuscript at line 168: "*The main goal of this study is to determine if the DCS can detect small $CH_4$ concentration enhancements downwind from the area of interest, equivalent to those caused by a typical herd of cattle grazing on an extensive pasture.*"  This is a reasonable objective, but it does not require using the DCS measurements to estimate emissions (using the IDM technique). Other verification trials have shown that IDM can accurately give emissions when provided an accurate concentration measurement. So the authors could drop the emission calculations to simplify the paper.

We moved the modified the objective to make the main goal of the paper more clear and included the statement in the introduction, specifically on lines 61 to 62 as

The main goal of this study is to determine if the dual-comb spectroscopy (DCS) combined with an IDM can precisely infer $CH_4$ fluxes from a typical herd of cattle grazing on an extensive pasture.

We kept the emission calculations because, as noted by the reviewer below, the DCS concentration measurements will be utilized with an IDM to infer trace gas fluxes. So, we believe it's important to show how the two techniques perform at estimating fluxes from a small herd as the one simulated in this study. This will make the results of the paper more appealing to a broader audience that is often more interested in the final product, namely emission estimates, than the concentration measurements.

2.  Despite the above comment, the paper is potentially more interesting when the DCS measurements are used to estimate emissions (with IDM). I would be OK making this task the main objective.  But the description of the tracer release verification needs improvement (comments below).

We have added more details to the emission estimates.

3.  In terms of dropping material … what is the value of the dispersion model calculations in the section "*Sensitivity and precision required for grazing measurements*"?  If it is to estimate concentration enhancement levels, can these instead be determined from the tracer release results (e.g., Fig 5, 7, 8)?  I prefer to see real-world measurements used for this task.  Another reason to drop the modelling work is that it is inadequately described (comments below).

We believe the forward modeling section is important to demonstrate how the methane plume concentration varies with distance from the herd. These simulations were also useful to test the placement of sensors in the field.  Furthermore, the scaling of the $CH_4$ precision in the section 'Sensitivity and precision required for grazing measurements'  provided us the minimum signal-to-noise ratio needed to obtain during the measurement to see the estimated small enhancements.  The controlled-release results confirmed these WindTrax simulations which we believe are important results to show.

We have improved the description of this section and lines 111 to 123

Wind orthogonal components and temperature data were measured at 10 Hz using a sonic anemometer (CSAT3, Campbell Sci, Logan, UT) deployed 5 m above a grazing unit on the Rannells' Flint Hills Prairie Preserve (full site description below) near Manhattan, Kansas USA. The wind dataset selected for these simulations consisted of about 30 days in June, 2021 during the grazing season. The wind raw data files were processed using the software Eddy Pro (Licor, Lincoln, NE) and means, variances and covariances for wind velocity and sonic temperature data were calculated for 30-min intervals to be used as input variables for the Windtrax simulations. To investigate if the DCS system can resolve the expected increase in $CH_4$ concentration due to the presence of cattle above the typical $CH_4$ atmospheric background level (2000 nmol mol-1), two expected $CH_4$ emission scenarios were evaluated: 100 g head-1 day-1 and 300 g head-1 day-1. These values were selected based on the reported IPCC Tier 1 emission values for grazing cattle in North America of 208 g head 1 day-1 (Eggleston et al., 2006). The simulated herd consisted of a fixed grid of point sources spaced 20 meters apart (Fig. 2). The height of gas release was set to 1 meter above the ground to mimic the height of the animal mouth and a total of 50,000 particles were released for each point source. Three beam lines were used in this simulation located at 45, 160 and 310 m from the geometric center of the herd.

.
Specific Comments
4.    Ln 57: "*In typical IDM applications, open-path Fourier Transform Infrared (FTIR) sensors are setup upwind and downwind for the source of interest*". I would not refer to use of an FTIR as the "typical" IDM application -- line-averaging lasers (TDLAS) have been more commonly used.  I suggest substituting "line-averaging" for "FTIR" here.  The association of IDM with "FTIR" also wrongly suggests the McGinn et al. (2011) study used an FTIR (they used a laser).

We thank the reviewer for pointing out this error.  We rewritten the statement in line 55 as

In typical IDM applications, open-path line-averaged concentration sensors are placed upwind and downwind from the source of interest.

5.    Ln 63: "*The limited path distance, bulky apparatus, multi-component retroreflectors (Bai et al., 2022) make employing open-path FTIR challenging in agricultural environments.*" Assuming the authors substitute "line-averaging" for "FTIR" as suggested above, they will have to either remove this list of FTIR disadvantages, or introduce the FTIR as a preferred (?) instrument in order to justify including this list.

 We have removed the sentence.

6.    Ln 112: "*A forward Lagrangian stochastic model (Windtrax, Thunderbeach Sci.) was used to simulate the concentration field downwind from a hypothetical herd of 20 head of beef cattle grazing in an area of 25 ha …*".  Give more details about the simulation: How were the point sources spatially distributed (randomly, evenly, moving)?  Where were the laser lines located relative to the sources (e.g., what does 45 m away from the herd mean)?  What was the `height of the release?  How many model particles were used?  Clarify these details.

We have added more details about the simulation in Section 'Sensitivity and precision required for grazing measurements'.  See Reviewer #1 general comment 3.

7.    Ln 114: "*Turbulence data for the simulations were measured … at a grazing unit adjacent to the measurement site. The wind dataset selected …*"  Explain which wind parameters were used.  Was the wind direction allowed to vary?

We have added more details about the simulation in Section 'Sensitivity and precision required for grazing measurements'. Yes, the wind direction was allowed to vary similarly. See Reviewer 1 comment 3 for the details on changes to the text.

8.    Ln 125: "*The forward model predicted that a herd of 20 cattle grazing in an area of 25 ha would produce a $CH_4$ enhancement of 16 nmol $mol_{-1}$ above a 2000 nmol $mol_{-1}$ background for a beamline 45 m away from the …*".  The text implies the concentration enhancements were calculated from 30 days of wind measurements (line 115).  Over time the enhancement will change as the wind conditions vary.  So what does the single enhancement value of 16 nmol

mol-1 represent (average value, median)?  What do the enhancement values in Table 1 represent?  What about the variability in the enhancement?

The enhancement refers to the average and the standard deviation has been added to table 1. We have also modified the table caption to explain the statistical parameters shown in the table.

Table 1 now reads:

Table 1 - Grazing system methane emission WindTrax simulation results showing the expected average and standard deviation (SD) of $CH_4$ concentration measured by line sensors positioned downwind from a cattle herd with two $CH_4$ emission rates. The $CH_4$ background level was assumed to be constant at 2000 nmol mol$^{-1}$.

| | Cattle $CH_4$ emission rate (g head$^{-1}$ day$^{-1}$) | | | | | |
|---|---|---|---|---|---|---|
| | 100 | | | 300 | | |
| Distance (m) | 45 | 160 | 310 | 45 | 160 | 310 |
| [$CH_4$] (nmol mol$^{-1}$) | 2005 | 2002 | 2001 | 2016 | 2006 | 2002 |
| SD [$CH_4$] (nmol mol$^{-1}$) | 12 | 4 | 2 | 36 | 12 | 7 |

9. Ln 162: This paragraph (or a variation of it) would be good in the introduction.  It gives important background information and contains a clear objectives statement: "*The main goal of this study is to determine if the DCS can detect small $CH_4$ concentration enhancements downwind from the area of interest, equivalent to those caused by a typical herd of cattle grazing on an extensive pasture.*"

We have moved the goal to the introduction and modified it following the suggestion made by Reviewer #1 in the general comments.

10. Ln 201: "*The $CH_4$ tank was weighted in the beginning and end of the gas release campaigns and the mass of gas released was determined gravimetrically …*"  For what purpose?  To verify the release rate given by the mass flow controller?  If so, did this confirm an accurate release rate?  Clarify.

We did not have a mass flow controller in this study. The manifold used in this study used a feedback loop to keep a constant gas delivery pressure during the controlled release. However, the apparatus in this study did not compensate for small temperature variabilities during the release campaigns. So, a gravimetric measurement would provide a more accurate measurement of the mass of gas released.  Line 210 now reads:

We used the mass given by the scale to determine the amount of gas released in each release campaign since it provides a more direct estimate of the release rate than the one obtained using the gas manifold. Previous gas release study has successfully used scale data to verify the flow rate of mass flow controller (Coates et al., 2017).

11.   Ln 220: "*The WindTrax input data consisted of … and appropriate wind statistics.*" Describe the wind statistics used.

In lines 225 and 226: We have replaced "appropriate statistics" by adding
mean, variance and covariances of wind velocity and temperature, obtained using the sonic anemometer data
.
12.   Ln 225: "*The source area (Fig. 4) used by WindTrax to infer fluxes was set to match the 12.5 $m_2$ area of the $CH_4$ point sources.*" Was the area source at ground level?

The source was set to 0.7m above the ground. Lines 231 now reads:

and the source level was set to 0.7 m above the ground which is the same height as the manifold outlets.

13.   Ln 233: "*… where F is the flux, $\sigma_F$ is flux error, $\sigma_{2rd}$ is downwind (background) mixing ratio error, $\sigma_{2ru}$ is upwind mixing ratio error, cov(d,u) is the covariance of the downwind and upwind errors.*" Are the $\sigma_2$ values variances (of what variable)?  Reference a good description of Eq. (2).

A reference for Eq 2 is the work of Herman et al., 2021,where this equation is based on sensitivity analysis found in Coddington et al. 2016.  The errors $\sigma_2$ are concentration variances, summed in quadrature to give the total error.  The text has been altered to fully explain the variables with proper references.

14.   Ln 241: "*A $CH_4$ release at a rate of 3078 g day-1 is shown in Fig. 5.*" It would be good to remind the reader of the significance of this release rate (e.g., it corresponds to X number of cattle)?

In line 247 we added:

A $CH_4$ release at a rate of 3078 g day-1 equivalent to 15 head of grazing cattle is shown in Fig. 5.

15.   Ln 244: "*This measurement demonstrates that the DCS system can detect small $CH_4$ enhancements equivalent to the ones caused by a small herd of cattle located at approximately 50 m from the downwind laser beamline.*"  A) Can this conclusion be justified statistically?  B) From Fig. 5 it appears there is no difference between $r_d$ and $r_u$ for some periods. The magnitude of ($r_d$-$r_u$) will depend on windspeed (i.e., u*), and plotting ($r_d$-$r_u$) vs u* would presumably show the DCS system is not detecting downwind enhancement for higher winds (when there is more dilution of the tracer).  This is not a unique problem for DCS as any sensor would show similar trends with u*.

We agree with Reviewer 1 that the statement in line 244 is not clear and the conclusion cannot be justified statistically. We have revised the statement to better describe the results in the Figure 4 and the impact of windspeed on the enhancement. Lines 251 to 255 now read:

However, the wind speed affected the DCS ability to measure these small concentration enhancements by diluting the methane plume as can be seen when the wind speed values were high during the afternoon of 04 Feb 2023 (Fig. 4a). The two-path DCS measurement was also capable of capturing the temporal dynamics of the $CH_4$ background driven by changes in atmospheric boundary layer conditions.

16. Ln 246: "*The two-DCS measurement path geometry is also capable of capturing and rejecting the temporal dynamics of the CH4 background driven by changes in atmospheric boundary layer conditions.*" What does "rejecting the temporal dynamics" mean?

We agree with Reviewer 1 that the phrase 'rejected' is unclear and it has been removed from the text.

17. Ln 252. "*To determine if any mixing ratio biases exist between the North and South beamlines that may lead to incorrect flux values… The average CH4 flux computed using WindTrax was ± 974 g day-1 equivalent to approximately 5 head of cattle …*". This paragraph is confusing and needs more explanation. A) Remind the reader of what you are doing, e.g., using path concentrations during a period with no gas release in order to determine precision of the DCS+IDM calculation? B) What is the difference between the given uncertainty in the average CH4 flux (+/- 217) and uncertainty using WindTrax (+/- 974)?

The purpose of this measurement is twofold: 1) to determine if there is any statistically significant bias between the north and south concentration measurements, and 2) use the no-release concentration result to provide minimum measurement sensitivity. Additionally, the statement is unclear and incomplete since the value of +/- 974 is the average flux error from Eq. 2. We have rewritten the statement to clarify the important results.

Lines 263 to 265 now read:

$CH_4$ dry mole fraction and WindTrax were used to compute an average $CH_4$ flux of 1.3 g day$^{-1}$ and standard deviation of ± 217.5 g day$^{-1}$. This standard deviation value is equivalent to approximately one head of cattle, assuming an emission rate of 200 g head$^{-1}$ day$^{-1}$.

18.    Ln 258. Figure 6 caption (and Fig. 7, 8).  Explain what the error-bars represent … flux uncertainties due to DCS measurement uncertainty (and this does not include IDM uncertainty).

The error bars are only uncertainties due to the DCS measurement since the IDM uncertainties are much smaller.  This has been clarified in the caption of Figure 6.

Figure 6: Time series of (a) wind friction velocity and direction, (b) dry $CH_4$ mole fraction  ($r_{CH4}$), and (c) release and IDM computed $CH_4$ flux a case of no released gas.  The error bars are uncertainties due to the DCS measured concentrations calculated using Eq. 2.  Wind arrows point in the direction from which the wind is blowing.

19.    Ln 266: "*WindTrax computed average CH4 flux was (4002 ± 1498) g day-1, showing a good agreement to the actual release CH4 flux of 3970 g day-1.*"  A) What does the ± value represent here?  B) The emission "recovery" fraction (4002/3970 = 1.008) is phenomenally good.  This is worth some commentary and context given other IDM tracer release studies (e.g., a compilation of verification studies is given in the appendix of Harper et al., 2010: The effect of biofuel production on swine farm methane and ammonia emissions.  J. Environ. Quality).

The flux value includes the average and the error computed by Equation 2.  We have removed the bracket notation and written the values explicitly out in the text.

Lines 275 to 277 now read:

WindTrax computed average $CH_4$ flux was 4002 g day$^{-1}$ and the flux uncertainty due to DCS concentration errors (Eq. 2) ± 1498 g day$^{-1}$, showing a good agreement to the actual release $CH_4$ flux of 3970 g day$^{-1}$.

We thank the reviewer for bringing attention to the 'recovery factor' and results published in Harper et al.  To address the comment we have added the following statement to the text.

As a point of comparison, Harper *et al.*, 2010 summarized the accuracy of IDM in 13 controlled release studies. They expressed the IDM accuracy by a recovered rate, given by ($F_{IDM}/F_{release}$)×100, finding an average recovery rate of 95% for all the studies. We estimated our recovery rate to be 100.8 (4002/3970

× 100) using the data shown in Fig. 7a. This is a noteworthy result indicating that the combination of DCS with IDM can produce flux estimates with high accuracy.

20. Ln 290: "*The agreement between the computed and actual CH4 fluxes in this study shows that DCS can precisely measure the small concentration enhancements due to a herd of beef cattle in the field at distances up to 100 m from the source area.*" A) It would be good to add the caveat about the effect of windspeed on detectability (see comment 15 above). B) I would like the authors to comment on the base level uncertainty (~ 5 cows), e.g., what does this imply about the minimum number of cattle that could be measured?

We have addressed this concern with the text added with respect to comment 15.

21. Ln 302: "*For example, prior calibrations are often necessary when using multiple FTIR systems to perform multi-path gas concentration measurements.*" What is meant by "prior calibrations"? Cross-calibration of different instruments? This is not just an FTIR problem, but a problem with all types of concentration sensors.

Cross calibrations will be necessary when using multiple instruments. We have clarified this point in lines 313 and 314, which now read:

For example, cross calibrations are often necessary when using multiple line average sensors to perform multi-path gas concentration measurements.

22. Line 305: "*Expected CH4 horizontal gradients in grazing systems are often small, as demonstrated in this study, so small instrument biases can lead to large errors when inferring fluxes.*" Good point.

Thank you for the comment.

23. Ln 310: "*The driving rationale of this work is to quantify the net CH4 fluxes produced by cattle grazing system …*" Are the authors suggesting a role for the DCS system in measuring soil fluxes (by either micrometeorological approaches or chambers)? Is this realistic given the generally small magnitude of the soil fluxes or the logistics of pairing DCS with a chamber.

No, we are not suggesting using the DCS system to measure small soil fluxes. The DCS measurements could be combined with other measurements, such as eddy covariance or soil chambers, to determine fluxes of soil and cattle components. We wrote in line 324:

Separating animal and soil contributions to the net $CH_4$ fluxes will require a combination of measuring approaches, such as chamber and micrometeorological measurements (e.g. eddy covariance measurements).

---

## Author Comment (AC2)

**Response to RC2's comments**

We would like to thank Reviewer 2 for the careful reading of our manuscript and the many comments below.  Below we address each comment specifically. The reviewer comments are written out in **black** text. Our responses are indicated in **blue** font, with text added to the manuscript indicated in **red** font.

The manuscript describes the field deployment of an dual comb spectroscopy (DCS) system along two simultaneously operated open measurement paths with the goal to measure CH4 emissions from cattle. This capability is tested and demonstrated in a controlled release experiment simulating cattle emissions while measuring along approximately 200m long paths upwind and downwind of the release area. The paper describes the experimental setup, hardware, the data processing, and spectral analysis which in the end produce path averaged mixing ratios. It further describes the analysis to infer flux estimations from these mixing ratios and their differences. This work provides a contribution in the ongoing challenge of measuring methane emissions from ruminants and on the open question of how open-path measurements of greenhouse gases (GHGs) are best employed in practice. I recommend publishing this work after addressing some minor and some potential major comments below.

General remarks:

I think you do not clearly and transparently present the extend of your measurement campaign (i.e. line 153 ff.). For what timespan was the DCS setup deployed? How many release experiments did you perform and on which days? The data you present in Figures 5 to 8 spans at least 3 months, yet you do not show for example the emission estimates for the day in Figure 5 which you used to show the enhancements during the release. Did you operate the open-path system during the mentioned grazing period (May to mit July) and try to measure real cattle emissions? If so, what were the additional challenges compared to your controlled release experiment? Since your stated goal is demonstrating the capability of such a system to monitor emissions, I would encourage you to state transparently how often you had high quality results. In its current state, the manuscript generates the impression that the extend of the dataset is intentionally vague and potentially data presented very selectively.

Our objective was to do a limited number of releases in the best wind conditions for the field beam geometry.  This was done in part to reduce the amount of methane we would release in the atmosphere. The number of days with favorable winds was smaller than what we expected based on past meteorology data for the site, so the overall time we were at the site was longer than expected.   DCS data were taken for those times, but since no cattle was present at the site only background emissions were measured and that data does not add anything to this manuscript.  The release data presented in the paper is nearly all the data measured, except for one run where an error in the gas manifold prevented us from getting the proper release rate.  Real cattle emissions were measured in the spring of 2023 where enhancements were measured.  However, our inability to properly track animal positions prevented us from calculating flux from these measurements, and thus the data are not presented here. Tracking animal positions is an additional challenge, one we have not solved at this point, to doing real grazing measurements.

Detailed remarks:

Line 23 f.: I do not see how the provided materials show that only optical power limits the measurement. The controlled release experiment had quite accurate knowledge of the release area and the manuscripts does not provide a systematic analysis of the impact of source distribution uncertainties and transport uncertainties,which typically contribute significantly to the uncertainties of fluxes estimated from concentration measurements.

The ability to detect small changes in molecular absorption (thus gas concentration) depends on the measurement signal-to-noise ratio (SNR), where the SNR depends on the optical power that gets back from the alignment between transceiver and retroreflector. If the SNR is poor, then there is more error in measuring small changes in the molecular absorption with respect to the background. This may not be clear in the abstract for a more general audience so the phrase was removed from the abstract.

Line 61 f.: To my knowledge, McGinn et al. (2011) did not use an FTIR system.

This has been fixed in the text.

Line 71: I assume with "square-law photodetector" you mean a photodiode operated in a linear (power to current) regime. If so, calling it that way might make this more accessible to a wider audience. If not, I do not understand the point you are making here.

The reviewer is correct on the meaning of "square-law photodetector", written this way to be more accessible to a wider audience.

Line 82 f.: I appreciate you citing LMFIT but think, if you do it, the doi should be included in some form: https://dx.doi.org/10.5281/zenodo.11813

The doi was somehow dropped by the code that generated the reference. This link has been added to the text.

Line 83: You did not cite the most recent version of HITRAN (HITRAN 2020, Gordon et al. 2022). If you did not use the most recent version, you might want to mention which version you used and your reasoning behind that. You also might want to mention which information (i.e. line shape model) you used.

We have updated the specific HITRAN versions used, added a reference to HITRAN Application Programming Interface (HAPI) including the Voigt lineshape used. That being said, we use HITRAN 2008 for methane which is trusted and tested for multiple outdoor

measurements done by NIST and our collaborators.

Line 108 f.: I think the concept of "molecular time" would be worth a one line explanation somewhere in the manuscript if you need it.

'Molecular time' refers to the picosecond timescale associated with the period of molecular oscillations.  This timeframe occurs too fast to be observed directly by a detector; the optical heterodyne maps these picosecond oscillations to microsecond oscillations which we detect with a fast photodetector.  A clearer one-line explanation has been added to Figure 1 caption.

'Molecular time' is the timescale associated with the period of molecular oscillations which is typically picoseconds.

Line 114: What does "Turbulence data" include? Which parameters were measured and are available for analysis?

We have addressed this comment see reviewer #1 specific comment #7.

Line 138 f.: Your estimation of your measurement sensitivity of 5 nmol/mol is really interesting and I think it would be worth a bit more thorough explanation. I for my part found it challenging to follow you and am still not certain if I understood it correctly.

The purpose of this discussion was to provide an estimate based on relationships between minimum gas concentrations and signal-to-noise-ratio described in Newbury et al., 2010. However, this discussion led to much confusion and relies heavily on the relations in the cited paper, which is not clearly expressed in that paper and is not a discussion  accessible to a wider audience.  Additionally this estimate does not add value to the paper since the actual precision is determined experimentally by the Allan-Werle analysis.  To make things clearer we have removed the estimate.

Line 190: It seems like you did not show or state your exact path lengths anywhere and not in Figure 4.
The one-way path distances were 202 m for the North beamline and 203 m for the south beamline. We have modified the caption of Fig. 4 to include this information.

Line 208: How did you calculate your mole fractions? What did you use to estimate your total airmass?
We generate the molecular lineshape using HITRAN molecular parameters and Voigt lineshape generated by HITRAN Application Programming Interface (HAPI).  The mole

fraction is an independent variable in that function and the molecular gas parameters include both self and 'air' line broadening factors. Thus by fitting to this lineshape using look-up tables and LMFIT, we directly get the mole fraction of the gas. To make that more clear, we have added a reference directly to the HAPI documentation.

Line 249: While this is not wrong in any way, I feel that using the combination of arrows and the convention to give the wind direction by where the wind is blowing from generates an unnecessary potential for misunderstanding.

We understand the possible confusion with the direction of the lines, but this convention is commonly done in meteorology and the convention is described in the caption.

Line 254 f.: I do not understand the difference between your estimate of 1 +- 217 g/day using the IDM and your WindTrax estimation of +-974 g/day. I thought, that WindTrax is your IDM (Line 220).

This statement is written incorrectly and has been corrected, as described above in response to Reviewer 1 comment 17.

Figure 8 b-d: I would again encourage you to show a bit more context around the presented data. If you have the data from 13:00 to 14:00 available, your plots here would provide a better impression on how well the signal can be distinguished from the background.

Unfortunately, we do not have earlier data for that particular day to show. The background $CH_4$ variability can be seen in Figure 5.

Line 291 ff.: As in Line 23 f. I do not see where you presented data on the transmitted power over time and how this is limiting your performance.

This statement needs more context. The limit is seen by performing Allan-Werle analysis of data sets with different signal-to-noise ratios, whose results are not presented here. However, getting any light back to the detector is a limitation of the measurement, especially in a windy environment on a prairie in Kansas. So we have reworded the statement, getting rid of SNR and focused on how to improve through better alignment.

Line 298 ff.: Again, I do not see where you discussed the technical aspects you list in point 1).

This was discussed in the experimental section around lines 170. We added more text to that section and the conclusion to make this more clear.

Line 335: You might want to mention how contributed to drafting/writing up the manuscript.

We added the phrase 'All authors contributed to the writing of this manuscript.'

---

## Author Comment (AC3)

**Editor review: Using Open-Path Dual-Comb Spectroscopy to Monitor Methane Emissions from Simulated Grazing Cattle**

Chinthaka Weerasekara et al AMT

The paper is appropriate for AMT but requires some minor and semi-major revisions before publication. The comments from 2 anonymous referees are valid and should be addressed. In addition I would like to add my editor's technical comments and corrections below:

L16, also L219 (Eq 1) and many other instances:  the usage of "mixing ratio" and "mole fraction" throughout the manuscript is not correct usage.  For a mixture of A and B, eg A=CH4 and B=CH4-free air, the mixing ratio is defined as A:B, and the mole fraction is A/(A+B). At 2 ppm levels the difference between the two is small and the names are often interchanged, if incorrectly so. But at higher levels it is significant. Eg the mixing ratio of O2 : air is 21:79 =0.26, the mole fraction of O2 in air  is 21/100 = 0.21.  In eq 1, if Xch4 is the mole fraction in whole (wet) air, X/(X-Xh2o) is corrected for the variable water content and referred to as the dry air mole fraction.  All usage of these terms should be searched, reviewed and corrected throughout the manuscript.

We thank the editor for clarifying the terminology.  This discrepancy is due to poor or confusing definitions in the literature.  We have replaced the term 'mixing ratio' with the term 'dry mole fraction'

L62. I believe McGinn used TDL instruments not FTIR, please also note anonymous reviewers comment on FTIR vs laser instruments. I agree to use open path as the descriptor, not FTIR.

We have corrected this mistake in Reviewer 1, Comment 4.

L76: replace "ideal" with "potentially valuable" – this paper is trying to show this to be true – "ideal" assumes that it is (and you do not need to write this paper...)

This phrase has been replaced.

L82: reference to Newville et al is not sufficient, the reader should be able to find the reference through a doi or similar.

The zotero bibliography editor dropped the doi, this has been added by hand to the references.

L96: IGMs (plural) not IGM's (possessive).  Please check for other cases.

The  possessives have been replaced.

Table 1: this would be easier to read with a vertical line after the 1st and 4th columns. Also if CH4 were given as enhancements, at 2000 nmol/mol, not as mole fractions.

The columns have been added to the table. Please see the response to Reviewer 1, Comment 9.

L140 - 145 . I have trouble to follow this calculation of SNR and detection limits on several levels – I request that it be completely rewritten.

This concern was addressed in response to a comment from Reviewer 2.

The purpose of this discussion was to provide an estimate based on relationships between minimum gas concentrations and signal-to-noise-ratio described in Newbury et al., 2010. However, this discussion led to much confusion and relies heavily on the relations in the cited paper, which is not clearly expressed in that paper and is not a discussion accessible to a wider audience. Additionally this estimate does not add value to the paper since the actual precision is determined experimentally by the Allan-Werle analysis. To make things clearer we have removed the estimate.

L141, what is meant by "normalised" here? Concentration/amount (and to what level), pathlength? What are the units?

See above comment. This originally was written as 'optical depth' . It has been removed.

L142 should read (1 – exp(-alpha.L) ) for absorption, the 1 is missing. This equals ~ alpha.L if alpha.L is small
Optical depth is alpha.L.concentation and dimensionless.

(eg (cm^2 molec^-1).cm.molec cm^-3). What units have you used here?
SNR is calculated in measured intensity or transmission spectra, not in optical depth, which is not linearly related except for weak absorption. They are only the same for weak lines. A given noise level corresponds to a much larger increment of concentration for a line that is already strongly absorbed in the background. It is linedepth:noise that matters for detection limit, not signal:noise
So I have trouble to interpret the calculation of 5 nmol/mol uncertainty or detection limit, especially in view of the L141 comment above – I don't know the pathlength or concentration which lead to the 0.03 "normalised" optical depths, and noise should be applied to the transmission spectrum, not optical depth. It makes a big difference if the 0.03 od is for 1 nmol/mol or for 2000 nmol/mol.
Finally, please state how you define detection limit – commonly this an amount equivalent to 3 x noise in the spectrum.

See above comment. The estimation has been removed.

L147: 12/18/2022 - please avoid this date format, it is ambiguous in an international journal. Although unambiguous in this instance, it is safer to use 18-Dec-2022 or 2022-12-18 format.

The date formatting has been fixed throughout the text.

L150: Figure 1 should be Figure 3, and 2=> 4. Please check all figure captions, numbers and cross references in the text.

All figure caption numbering has been checked in the text.

L173-174 … that were used …  (not was used)

The tense has been fixed.

L183, 185, 189 : (PT100, FLIR  etc ) is not sufficient to identify the supplier.  Normal usage is model number, manufacturer and location, so they can be followed up.

The manufacturer location has been added where missing.

L219  see L16 comment

This has been fixed.

243: Would be better expressed as "Data from a CH4 release ….".  The figure referenc ed is also incorrect on this line.

The phrase has been added and the figure reference number has been checked.

L254-259. I cannot see how the last sentence in this paragraph relates to what comes before it. If the measured bias from the up-down measurement is 1 +/- 217 g /day using "the IDM", where does the 974 g/day using Windtrax come from?

This concern has been addressed in response to Reviewer 1, Comment 17.

---

## Author Response (AR2)

**Dear Editor:**

**Thank you for the detailed comments on the final version of our manuscript. You suggested edits have been made and included in the final version of the paper.**

**Sincerely,**

**Brian Washburn**

**Eduardo Santos**

**Public justification (visible to the public if the article is accepted and published)**:
Thank you for revising the manuscript after consideration of reviewer comments. These revisions have largely satisfied the reviewers' concerns and further major revision is not required. However on now reading the revised MS end to end, I would like to recommend some reorganisation of existing paragraphs for improved readability. I find the current breakup into (unnumbered) sections to be a little illogical – general methods are mixed up with the specifics of the field experiment, and Windtrax must be explained twice (once for simulations and once for flux estimations).

**The reorganization has been completed as suggested in the final version. We have added better numbering to the sections to help improve readability.**

I have made some explicit suggestions in the additional private note section

Additional private note (visible to authors and reviewers only):
In principle the general experimental and method details including DCS should all appear together in a Methods section (lines 64 – 108, 140-150, 210-220) followed by Windtrax and the IDM simulations (currently "Sensitivity and precision required for grazing measurements, L109-140 and "Computing CH4 flux…", line 223 - 244), then the specifics of the field measurements and results ("Controlled release experiment lines 152 – 300 after moving 210-220 and 223-244), and then summary and conclusions.

**Thank you for the suggestions, this will make the paper clearer. All information has been added in the 'Methods' section as suggested. See the marked up version of the manuscript for specific changes.**

After rearranging sections/paragraphs, please check for logical consistency and flow, so the general methods precede their specific application.

I also recommend sections and subsections, preferably numbered, it is then much easier for cross referencing. Suggestion for logical flow:
1. Introduction and motivation
2. Methods
a. DCS
b. Obtaining CH4 mole fractions using spectral line fitting
c. IDM, Windtrax
i. Simulations of sensitivity and precision
ii. Computing CH4 flux using an inverse dispersion model
3. Controlled CH4 release experiment
a. Description
b. Results
4. Future work and conclusions

**The paper structure has been rearranged as suggested.**

Minor technical corrections:
L185 Campbell is mis-spelt
L241 sigma-squared-rd appears twice, one should be ru

**These minor corrections have been made.**